# A preclinical model of THC edibles that produces high-dose cannabimimetic responses

Anthony English[1,2,3], Fleur Uittenbogaard[1,2,3], Alexa Torrens[4], Dennis Sarroza[1], Anna Veronica Elizabeth Slaven[1,2], Daniele Piomelli[4], Michael R Bruchas[1,2,3,4,5], Nephi Stella[1,2,3,6]*, Benjamin Bruce Land[1,2,3]*

[1]Departments of Pharmacology, University of Washington, Seattle, United States; [2]UW Center of Excellence in Neurobiology of Addiction, Pain, and Emotion (NAPE), University of Washington, Seattle, United States; [3]Center for Cannabis Research, University of Washington, Seattle, United States; [4]Department of Anatomy & Neurobiology, University of California Irvine, Irvine, United States; [5]Department of Anesthesiology, University of Washington, Seattle, United States; [6]Psychiatry & Behavioral Sciences, University of Washington, Seattle, United States

*For correspondence:
nstella@uw.edu (NS);
bbl2@uw.edu (BBruceL)

**Competing interest:** The authors declare that no competing interests exist.

**Abstract** No preclinical experimental approach enables the study of voluntary oral consumption of high-concentration $\Delta^9$-tetrahydrocannabinol (THC) and its intoxicating effects, mainly owing to the aversive response of rodents to THC that limits intake. Here, we developed a palatable THC formulation and an optimized access paradigm in mice to drive voluntary consumption. THC was formulated in chocolate gelatin (THC-E-gel). Adult male and female mice were allowed ad libitum access for 1 and 2 hr. Cannabimimetic responses (hypolocomotion, analgesia, and hypothermia) were measured following access. Levels of THC and its metabolites were measured in blood and brain tissue. Acute acoustic startle responses were measured to investigate THC-induced psychotomimetic behavior. When allowed access for 2 hr to THC-E-gel on the second day of a 3-day exposure paradigm, adult mice consumed up to ≈30 mg/kg over 2 hr, which resulted in robust cannabimimetic behavioral responses (hypolocomotion, analgesia, and hypothermia). Consumption of the same gelatin decreased on the following third day of exposure. Pharmacokinetic analysis shows that THC-E-gel consumption led to parallel accumulation of THC and its psychoactive metabolite, 11-OH-THC, in the brain, a profile that contrasts with the known rapid decline in brain 11-OH-THC levels following THC intraperitoneal (i.p.) injections. THC-E-gel consumption increased the acoustic startle response in males but not in females, demonstrating a sex-dependent effect of consumption. Thus, while voluntary consumption of THC-E-gel triggered equivalent cannabimimetic responses in male and female mice, it potentiated acoustic startle responses preferentially in males. We built a dose-prediction model that included cannabimimetic behavioral responses elicited by i.p. versus THC-E-gel to test the accuracy and generalizability of this experimental approach and found that it closely predicted the measured acoustic startle results in males and females. In summary, THC-E-gel offers a robust preclinical experimental approach to study cannabimimetic responses triggered by voluntary consumption in mice, including sex-dependent psychotomimetic responses.

## eLife assessment

This **important** study presents the validation of an oral $\Delta^9$-tetrahydrocannabinol (THC) consumption mouse model utilizing highly palatable e-capsule gelatin. The results **convincingly** demonstrate that oral consumption produced THC behavioral and physiological effects, as well as measurable brain

levels. The utility of the model for chronic consumption remains to be determined. The authors have clearly acknowledged limitations of their model and areas for future study and development. As the field of cannabinoid research moves toward application of routes of administration that mimic human use, these model systems will be increasingly **important** in assessing the effects of cannabinoid-based drugs.

## Introduction

The use of *Cannabis* products containing high concentrations of $\Delta^9$-tetrahydrocannabinol (THC) is rapidly increasing despite our limited understanding of its potential impact on physical and mental health (*Hasin et al., 2019*; *Compton et al., 2016*; *Carlini and Schauer, 2022*). These products are typically inhaled as combusted plant matter, vaporized extracts, or consumed in edible formulations. THC acts as a partial agonist at cannabinoid 1 receptors ($CB_1R$) to trigger a myriad of responses such as physiological responses (e.g., increase heart rate), altered mood and time perception, inhibition of motor control, and impaired learning and memory (*Martin-Santos et al., 2012*; *Zuurman et al., 2008*; *Morgan et al., 2018*; *Hollister and Gillespie, 1973*; *Hollister, 1970*; *Weinstein et al., 2008a*; *Weinstein et al., 2008b*). Subsequently, a relationship between *Cannabis* and psychotic/affective symptoms and an observable increase in *Cannabis*-associated vehicle crashes has become apparent without an understanding of the neural effects of higher-dose THC (*Morgan et al., 2018*; *Hindley et al., 2020*; *Barrett et al., 2018*; *Lira et al., 2021*; *Moore et al., 2007*). Many of these effects translate to preclinical models, where in rodents THC reduces spontaneous locomotion and locomotor control, and induces hypersensitivity to tactile and auditory stimuli, ataxia, and sedation; all of which have been shown to be mediated through action at $CB_1R$ (*Holtzman et al., 1969*; *Beardsley et al., 1987*; *Metna-Laurent et al., 2017*; *Siemens and Doyle, 1979*). Importantly, some cannabimimetic responses are sex-dependent, as exemplified by the finding that THC (5 mg/kg, intraperitoneal [i.p.]) triggers a more pronounced reduction in spontaneous locomotion and anxiogenic response in females than in males (*Ruiz et al., 2021*; *Kasten et al., 2019*). In addition to these cannabimimetic responses, preclinical investigations have pursued psychosis-related behaviors through the acoustic startle response, finding that involuntary administration of THC impairs psychomotor/sensorimotor gaiting (*Ibarra-Lecue et al., 2018*; *Long et al., 2010*; *Boucher et al., 2011*; *Hoffman and Ison, 1980*), emphasizing the translational value in understanding THC's bioactivity in humans.

Understanding the effects of increased THC use in humans through preclinical models of voluntary THC administration has proven difficult to establish due to the aversive behaviors to higher doses in rodent models (*Burgdorf et al., 2020*; *Lepore et al., 1995*). In recent years, progress has been made in promoting voluntary oral consumption of THC in rodents, but results have been limited to mild, acute $CB_1R$-dependent cannabimimetic responses (*Kruse et al., 2019*; *Abraham et al., 2020*; *Smoker et al., 2019*). This lack of experimental tools to translate higher-dose THC intake in humans to preclinical models emphasizes the urgent need to develop and fully characterize a novel experimental approach. To bridge this translational gap, we initially developed an approach where mice are given ad libitum access to consume a sugar-water gelatin (CTR-gel) containing fixed amounts of THC (*Abraham et al., 2020*; *Kruse et al., 2019*). Matching previous rodent studies, we found that mice consumed more vehicle gelatin than THC gelatin, indicating that they detected and avoided THC (*Barrus et al., 2018*). To overcome this limitation, in this study we developed and characterized a palatable oral gelatin formulation that increases voluntary consumption by formulating THC in a chocolate-flavored nutritional shake, Ensure (E-gel). Previous work has shown that mice have a preference for chocolate flavor, making it an ideal THC formulant to increase palatability (*Ventura et al., 2007*; *Barbano et al., 2009*). We leveraged this approach to determine whether oral consumption of high-concentration THC gelatin induces commonly studied cannabimimetic responses in mice (hypolocomotion, analgesia, and hypothermia), and then we examined the effects of THC E-gel consumption on acoustic startle response, a preclinical measure of reflexive response rate and psychomotor arousal (*Long et al., 2010*; *Nagai et al., 2006*). The model developed here leverages acute voluntary consumption of a sweetened gelatin to investigate psychomotor and reflexive behaviors, pharmacokinetics, and triad responses following consumption of THC E-gel in mice.

## Materials and methods

### Animal studies

Animal studies followed the guidelines established by the Association for Assessment and Accreditation of Laboratory Animal Care (AAALAC) and were approved by the Institutional Animal Care and Use Committee (IACUC) of the University of Washington. Male and female C57BL/6J mice ranging from 8 to 14 wk of age were used. Animals were housed with sibling littermates and were provided with standard chow and water, ad libitum, and without any additional environmental enrichment. Investigators were not blinded to experimental exposure conditions throughout assays due to the noticeable behavioral effects measured in response to THC. Animal procedures were approved by the IACUC of the University of Washington and conform to the guidelines on the care and use of animals by the National Institutes of Health.

### Pharmacological agents

Animals received THC (0.1, 0.3, 1, 3, 5, 10, and 30 mg/kg) and SR141716 (SR1, 1 mg/kg) i.p. or were exposed to THC suspended in gelatin. THC and SR141716 (SR1) were provided by the National Institute of Drugs Abuse Drug Supply Program (Bethesda, MD). THC in ethanol (50 mg/ml) was either added to gelatin mixtures (CTR or Ensure) or prepared for i.p. injection. For i.p. injection, both THC (0.1, 0.3, 1, 3, 5, 10, and 30 mg/kg) and SR1 (1 mg/kg) were dissolved in 95% ethanol and then vortexed thoroughly with equal volume cremophor and finally dissolved in sterile saline to reach a final 1:1:18 solution consisting of ethanol:cremophor:saline.

### Gelatin formulation

#### Control gelatin (CTR-gel)

Deionized water (100 ml) was warmed to 40°C and stirred at a constant rate. 2.5 g of Polycal sugar and 3.85 g of Knox gelatin was added, and the mixture was maintained at a temperature below 43°C. The mixture was removed from heat, and THC (50 mg/ml in ethanol) was added to reach a concentration of 0.3, 1, 2, or 4 mg/15 ml. An equal volume of ethanol was added to vehicle gelatin (<1% total volume). Gelatin was poured into plastic cups ranging from 2 to 10 ml and set into a 4°C fridge to solidify overnight.

#### Ensure gelatin (E-gel)

Chocolate-flavored Ensure (100 ml) was warmed to 40°C and stirred at a constant rate. 3.85 g of Knox gelatin were added, and the mixture was maintained at a temperature below 43°C. The mixture was removed from heat, and THC (50 mg/15 ml ethanol) was added to reach a concentration of 1, 2, 5, or 10 mg/15 ml. At the 10 mg/15 ml concentration, ethanol was evaporated off to 50% volume before being added to the mixture to reduce total alcohol concentration below 1%. An equal volume of ethanol was added to vehicle gelatin (<1% total volume). Gelatin was poured into plastic cups ranging from 2 to 10 ml and set into a 4°C fridge to solidify overnight. Mice were always exposed to more gelatin than they could consume, and smaller volumes were used to conserve THC.

#### Acute gelatin access

Animals were first habituated to gelatin by receiving an excess of gelatin in their home cage the day before the first timed access. On the first day of access, mice were placed into a home cage-like environment equipped with a vehicle gelatin cup that was stabilized in the cage. Behavior was recorded during the consumption window via an overhead camera. On the second day of access, animals were placed into the same gelatin access cage with either a vehicle or THC gelatin cup. Animals experienced either a triad of behaviors (open field, tail flick, and body temperature) measured immediately preceding and following the consumption window or an acoustic startle trial immediately following consumption. On the third and final day of access, animals were placed into the same cage with a vehicle gelatin cup. For all gelatin access days, gelatin cups and animals were weighed before and after the consumption window. Access to gelatin during the consumption window was limited to either 1 or 2 hr after which the animals were removed and returned to their home cage.

### Triad of cannabimimetic behaviors

Hypolocomotion, hypothermia, and analgesia were measured 1 hr post-i.p. injection or immediately following gelatin exposure. Pre-tests were collected immediately prior to injection or gelatin exposure.

## Open field

A 50 cm × 50 cm chamber (25 LUX) was equipped with an overhead camera to record movement. Animals were placed in the chamber for 15 min and then returned to their home cage. Total distance traveled (cm) was measured using Noldus Ethovision behavioral tracking software. Locomotion behavior was measured immediately before and after gelatin access to calculate a gelatin-dependent difference score (post-pre). Pre-test measurements for CTR-gel were not collected, and pre-test values were instead normalized to vehicle post-tests to produce a difference score as Post-VEH.

## Tail flick analgesia

A hot water bath was set to 52.5°C. Mice were securely held upright in the air with their tail hanging downward. A timer was started as 75% of their tail was submerged in the water. Time was measured once a painful response was presented, marked as a latency to flick their tail out of the hot water. Tail flick responses were measured immediately before and after gelatin access to calculate a gelatin-dependent difference score (post-pre).

## Measuring body temperature

Animals were placed on a stable surface with their tails lifted. A rectal thermometer probe (RET-3 Kent Scientific) was inserted into the anus for 10–20 s until the temperature recording stabilized. This test was always performed prior to the tail flick test to reduce any potential temperature contamination effects. Body temperatures were measured immediately before and after gelatin access to calculate a gelatin-dependent difference score (post-pre).

## Blood and brain tissue collection and quantification

Animals underwent the same gelatin access paradigm for days 1 and 2 described in acute gelatin access. After 2 hr of gelatin access, blood was collected by cardiac puncture with a 23-gauge needle and placed on ice. Immediately following, brain tissue was collected and flash-frozen in liquid nitrogen. Blood samples were spun in a 4°C centrifuge at 1450 × *g* for 15 min. Plasma was transferred to another tube and stored alongside brain samples at –80°C until being shipped on dry ice to the Piomelli Lab at UCI for sample analysis. Samples were collected immediately following 1 and 2 hr gelatin access and 30 min and 24 hr after 2 hr gelatin access. THC and its first-pass metabolites 11-OH-THC and 11-COOH-THC were quantified in plasma and brain tissue using a selective isotope-dilution liquid chromatography/tandem mass spectrometry assay. Concentration data after E-gel consumption were compared to blood and brain tissue concentrations after i.p. administration from previous publications from the Piomelli Lab (*Vozella et al., 2019*).

## Acoustic startle

Acoustic startle behaviors were measured after 1 or 2 hr THC-E-gel exposure (10 mg/15 ml) and THC-i.p. (0.1, 1, 5, and 10 mg/kg) injection. Sound-buffered startle chambers (SR-Lab, San Diego Instruments) were used to measure acoustic startle responses, equipped with a holding tube and an accelerometer. Background sound was maintained at 65 dB from a high-frequency speaker producing white noise. Startle tests were conducted 1 hr post-THC-i.p. injection or immediately following THC-E-gel exposure. Animals were set in the holding tube for 5 min to habituate prior to a series of seven trials presenting escalating sound levels of null, 80, 90, 100, 105, 110, and 120 dB. Tones were presented for 40 ms with an inter-trial interval of 30 s. Animals were only ever exposed to the acoustic startle paradigm once, immediately after gelatin access, to avoid auditory habituation.

## Data/statistical analysis

All data were analyzed using GraphPad Prism 10-11. For all statistical analyses (unpaired *t*-test, one- and two-way ANOVA, and post hoc analyses), alpha level was set to 0.05. For all ANOVAs, Sidak's post-test was performed for increased power and repeated measures analysis was performed for time-dependent consumption data. All behavioral locomotor tracking was analyzed using Noldus Ethovision software, and statistical analyses were performed through GraphPad Prism 10-11. Nonlinear regressions (*Figure 1—figure supplement 3b–g*) were performed with a three-parameter nonlinear, least squares, regression.

# Results

## E-gel promotes heightened voluntary oral consumption of THC and induces cannabimimetic behaviors in adult mice

To incentivize voluntary oral consumption of high-concentration THC, we utilized an E-gel formulation and optimized an exposure paradigm based on previous studies (**Abraham et al., 2020**; **Schindler et al., 2014**). Here, individual mice were exposed to a control (CTR-gel) during a 2 hr consumption period (Habituation, day 1); and the following day exposed to THC formulated in either CTR-gel or E-gel (*X* mg/15 ml) for 2 hr (**Figure 1a**). Gelatin mass was measured before and after access to calculate grams consumed, and gelatin concentrations are expressed as *X* mg of THC (*X* = mg of THC/15 ml gelatin) (**Figure 1b**). As expected, higher concentrations of THC-CTR-gel reduced gelatin consumption, an effect significant at 1 mg THC (**Figure 1c**, one-way ANOVA $F_{4,108}$ = 9.126, p<0.001, Sidak's). Remarkably, 7/17 (41%) of mice exposed to 4 mg CTR-gel did not consume any gelatin while all mice consistently consumed 5 mg and 10 mg THC-E-gel (**Figure 1d**). Thus, mice consumed 1.9 ± 0.05 g of VEH-E-gel and 1.0 ± 0.07 g of E-gel containing THC (10 mg/15 ml) (**Figure 1d**, one-way ANOVA $F_{5,75}$ = 14.10, p<0.001, Sidak's). Note that mice consumed similar amounts of VEH-E-gel and VEH-CTR-gel (1.96 ± 0.15 g and 1.92 ± 0.17 g, respectively), indicating that chocolate flavor per se does not increase consumption. When calculating the amount of THC consumed in mg/kg, we found that mice consumed more THC when formulated in E-gel (**Figure 1e**). For example, mice consumed 10.5 ± 0.7 mg/kg/2 hr when exposed to 2 mg E-gel compared to only 4.6 ± 0.5 mg/kg/2 hr when exposed to the same amount of THC formulated in CTR-gel. Using this experimental approach, maximal consumption reached 29.2 ± 1.8 mg of THC per kg/2 hr when exposed to E-gel containing THC (10 mg/15 ml), compared to the few mice that consumed only 8.4 ± 1.2 mg of THC per kg/2 hr when exposed to CTR-gel containing THC (4 mg) (**Figure 1e**, two-way ANOVA $F_{7,94}$ = 73.14, p<0.001, Sidak's). As previously shown, we found no statistically significant sex-dependent effects in consumption between male and female mice across all treatments (two-way ANOVA $F_{1,60}$ = 3.64, p<0.06) but did see an individual significance between males and females at 1 mg THC-E-gel (**Figure 1—figure supplement 1**, two-way ANOVA, p<0.05; **Abraham et al., 2020**; **Kruse et al., 2019**). These data show that mice consistently consume significant quantities of E-gel despite high THC concentrations. Based on these results, we next focused our study on quantifying the pharmacological effects of THC formulated in E-gel.

Considering that THC-CTR-gel triggers mild cannabimimetic responses due to limited consumption (**Abraham et al., 2020**; **Kruse et al., 2019**), we determined whether consumption of THC-E-gel could induce cannabimimetic responses measured immediately following the 2 hr access period (**Figure 1f**). Thus, we selected three well-described behavioral effects of THC in mice: hypolocomotion, analgesia, and hypothermia (**Figure 1g–i**; **Metna-Laurent et al., 2017**). THC was formulated and administered either by i.p. injection (gray), CTR-gel (green), or E-gel (purple), and behavioral responses to gelatin consumption were plotted from the average dose consumed, calculated in **Figure 1e**. **Figure 1g** shows that i.p. administration of THC reduced locomotion starting at 3 mg/kg, as previously reported (**Metna-Laurent et al., 2017**), and that this response was significant after access to 4 mg THC-CTR-gel (avg: 8.4 mg/kg) and 2 mg THC-E-gel (avg: 10.5 mg/kg) (**Figure 1—figure supplement 2b and e**, one-way ANOVA $F_{5,68}$ = 14.54, p<0.001, Sidak's for CTR-gel and $F_{4,56}$ = 3.24, p=0.02 for E-gel, Sidak's). **Figure 1h** shows the greatest THC-induced analgesia was reached at 30 mg/kg i.p. and after access to 2 mg THC-E-gel (**Figure 1h**, **Figure 1—figure supplement 2d and g**, one-way ANOVA $F_{4,104}$ = 9.4, p<0.001, Sidak's for CTR-gel and $F_{7,77}$ = 8.7, p<0.001 for E-gel, Sidak's). THC injection (i.p.) reduced core body temperature starting at 3 mg/kg, and that this response reached significance at 4 mg THC-CTR-gel (avg: 8.4 mg/kg) and at 5 mg THC-E-gel (avg: 17.2 mg/kg). **Figure 1i** also shows that reduced core body temperature induced by THC reached a significant effect of –5.84°C at 30 mg/kg i.p., –1.5°C after 4 mg THC-CTR-gel (avg: 8.4 mg/kg), and –1.8°C after 10 mg THC-E-gel (avg: 29.2 mg/kg) (**Figure 1—figure supplement 2d and g**, one-way ANOVA $F_{5,61}$ = 11.16, p<0.001, Sidak's for CTR-gel and $F_{4,114}$ = 6.36, p<0.001 for E-gel, Sidak's). Analgesia and hypothermia did not plateau, matching prior studies that have also shown that 30 mg/kg THC-i.p. does not produce a maximal response for these cannabimimetic behaviors (**Falenski et al., 2010**; **Varvel et al., 2005**). Sexual dimorphic responses were measured after THC administration through either i.p. injection or E-gel, although VEH-E-gel did show a significant difference in hypolocomotion between males and females (**Figure 1—figure supplement 3b-g**, THC: two-way ANOVA, Sidak's post-test, VEH-E-gel locomotion:

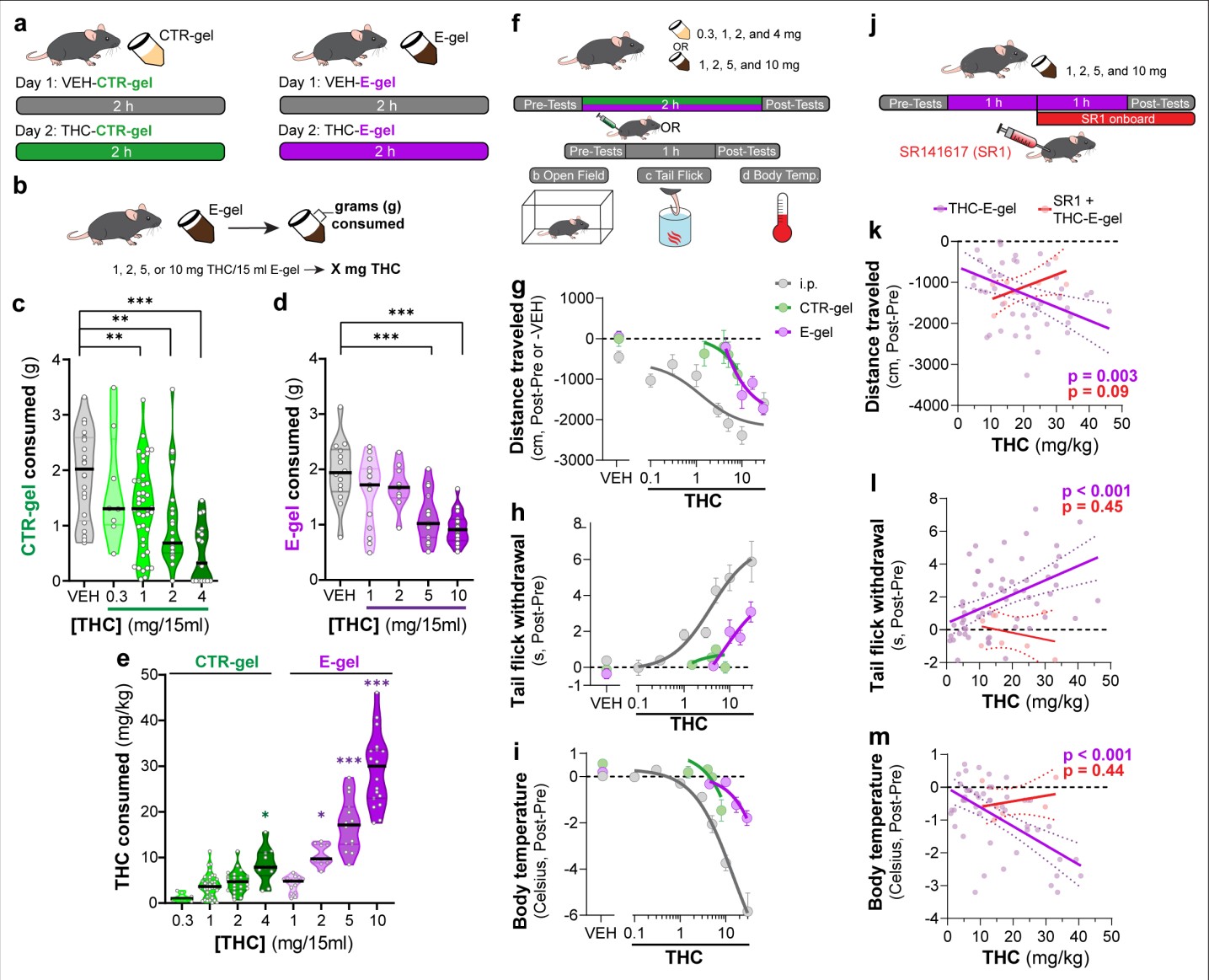

**Figure 1.** E-gel promotes heightened voluntary oral consumption of Δ⁹-tetrahydrocannabinol (THC) and induces cannabimimetic behaviors in adult mice. (**a**) Mice were given free access to vehicle (VEH) or THC formulated in either CTR-gel or E-gel for 2 hr on days 1 and 2. (**b**) Consumption was determined by weighing gelatin at the end of each session. (**c**) Consumption of CTR-gel on day 2 is decreased after addition of THC. (**d**) Consumption of E-gel on day 2 is maintained after addition of THC. (**e**) Dose of THC consumed, in mg/kg, when formulated in either CTR-gel or E-gel on day 2. Results are mean ± SEM. Consumption compared ANOVA and Sidak's, *p<0.05, **p<0.01, and ***p<0.001, N = 8–40. (**f**) Diagram of behavioral paradigm before and after intraperitoneal (i.p.) or gelatin administration. (**g–i**) Dose-dependent behavioral responses for hypolocomotion (**g**), analgesia (**h**), and hypothermia (**i**) after THC exposure. Administration by i.p. (gray) is plotted on x-axis by single bolus injection while CTR-gel (green) and E-gel (purple) are plotted based on average THC consumed after 2 hr exposure window shown in (**e**). (**j**) Diagram of THC-E-gel exposure, behavioral measurements, and SR1 injection (by i.p.) at 1 hr into exposure window. (**k–m**) Individual behavioral responses for hypolocomotion (**k**), analgesia (**l**), and hypothermia (**m**) for each animal. Individual points are plotted based on individual THC consumption with a linear regression to show correlation between consumed THC and behavioral output (p-values: *k* = 0.003, *l* < 0.001, *m* < 0.001). SR1-treated mice are plotted (red) based on consumed THC after exposure to 10 mg/15 ml THC-E-gel with a linear regression to show no correlation across three behaviors (p-values: *k* = 0.09, *l* = 0.44, *m* = 0.45).

The online version of this article includes the following figure supplement(s) for figure 1:

**Figure supplement 1.** CTR-gel and E-gel consumption by male and female mice.

**Figure supplement 2.** Triad behavioral responses after CTR-gel and E-gel consumption.

**Figure supplement 3.** Triad behavioral responses to Δ⁹-tetrahydrocannabinol (THC) intraperitoneal (i.p.) and E-gel administration by male and female mice.

unpaired *t*-test, $t = 2.721$, df = 16, p=0.02). Thus, *Figure 1g–i* shows that (1) THC reduces locomotion when administered using these three experimental paradigms, and to a greater extent at higher-dose i.p. THC and higher-concentration THC-E-gel; (2) THC induced analgesia only when administered i.p. and using THC-E-gel, though i.p. administration is more potent; and (3) THC reduces core body temperature only when administered i.p. and using THC-E-gel, though i.p. administration is more potent.

Next, we analyzed the cannabimimetic responses of individual mice following THC (10 mg/15 ml) E-gel access and how the $CB_1R$ inverse agonist, SR1, administered 1 hr into the consumption window influences these responses (*Figure 1j*). SR1 was administered at 1 hr to reach peak circulating concentrations during the behavioral testing (1–2 hr post-injection) and reduce any anorectic effects that would inhibit consumption of THC-E-gel (*Ettaro et al., 2020; Wright and Rodgers, 2013*). Cannabimimetic responses increased as a function (linear regression) of increasing amount of THC consumed, demonstrating a significant relationship between the amount of THC consumed and the three cannabimimetic readouts: hypolocomotion (linear regression $F_{1,46} = 9.74$, p=0.003), analgesia (linear regression $F_{1,57}=15.73$, p<0.001), and hypothermia (linear regression $F_{1,46} = 24.72$, p<0.001) (*Figure 1k–m*). Confirming the involvement of $CB_1R$, SR1 blocked the three THC-induced cannabimimetic responses: hypolocomotion ($F_{1,6} = 4.1$, p=0.09), analgesia ($F_{1,6} = 0.66$, p=0.45), and hypothermia (SR1: $F_{1,6} = 0.68$, p=0.44) (*Figure 1k–m*). As expected, SR1-treated mice did not consume maximal THC-E-gel compared to some animals exposed to 10 mg THC-E-gel, likely due in part to the SR1 injection 1 hr into the consumption window and the known anorectic effects of SR1 (*Ettaro et al., 2020; Wright and Rodgers, 2013*). We additionally compared the linear regression of all THC-E-gel consumption within the range of SR1-treated animal consumption and found that THC-E-gel alone still produced a significant correlation to all behaviors. These results indicate that consumption of THC-E-gel evokes robust $CB_1R$-dependent cannabimimetic behavioral responses in adult mice that are comparable to i.p.-THC administration when measuring hypolocomotion, and less potent when compared to i.p.-THC administration when measuring analgesia and reduction in core body temperature.

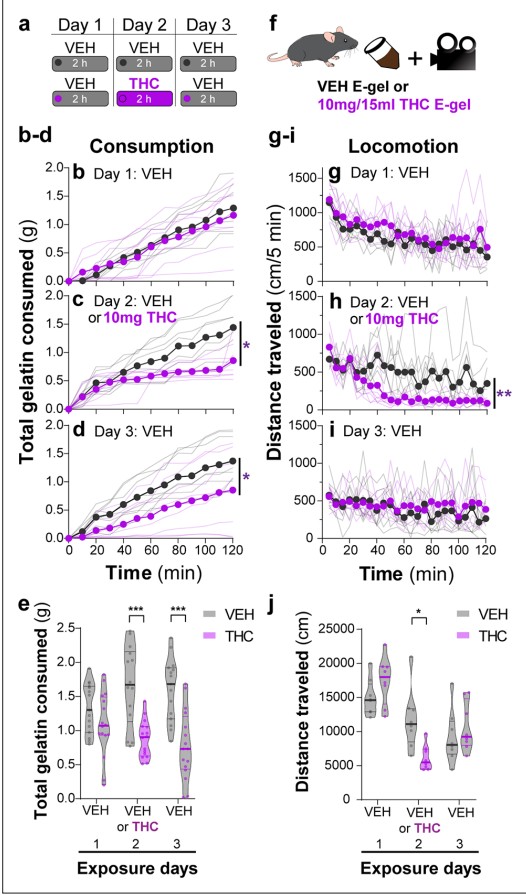

**Figure 2.** THC-E-gel consumption triggers $CB_1R$-dependent behaviors. (**a**) Over a 3-day exposure paradigm, mice received 3 d of E-gel with either vehicle (VEH) or $\Delta^9$-tetrahydrocannabinol (THC) (10 mg/15 ml) E-gel on day 2. (**b–d**) Cumulative gelatin consumption recorded every 10 min throughout the 2 hr exposure window over the 3-day paradigm. VEH (black) and THC (purple) groups received access to VEH on day 1 (**b**), VEH or THC on day 2 (**c**), and VEH on day 3 (**d**). (**e**) Total gelatin consumption after 2 hr of access to gelatin was plotted comparing VEH and THC treatment groups. (**f**) Animal consummatory and locomotor behavior was tracked during gelatin exposure window. (**g–i**) Distance traveled recorded every 5 min over the 3-day paradigm, similar to (**b–d**). (**j**) Total distance traveled (cm) after 2 hr of gelatin access was plotted comparing VEH and THC groups. Main effect over 2 hr exposure period (**b–d, g–i**) measured using two-way ANOVA with repeated measures and Sidak's, main effect on total response (**e, h**) measured by one-way ANOVA and Sidak's (*p<0.05, **p<0.01, ***p<0.001), N = 8–16.

The online version of this article includes the following figure supplement(s) for figure 2:

**Figure supplement 1.** Analysis of $\Delta^9$-tetrahydrocannabinol (THC) E-gel consumption behavior over 3-day access paradigm.

## THC-E-gel reduces locomotion during the exposure period

We found that mice consumed ~2 g of vehicle E-gel (VEH-E-gel) compared to ~1 g of THC-E-gel (10 mg/15 ml), indicating a twofold reduction in consumption (*Figure 2a*). To investigate the time course of this effect, we weighed gelatin every 10 min during the 2 hr access period in a 3-day paradigm: access to VEH-E-gel on day 1, access to either VEH-E-gel or THC-E-gel day 2, and access to VEH-E-gel on day 3 (*Figure 2a*). *Figure 2b* shows that consumption of VEH-E-gel on day 1 started within 20 min of availability and was constant during the 2 hr period. On day 2, mice consumed comparable amounts of VEH-E-gel and THC-E-gel during the initial 40 min of access (16.3 and 13.0 mg/min, respectively) (*Figure 2c*, *Figure 2—figure supplement 1a*). However, consumption of THC-E-gel plateaued after 40 min to a rate of 4.2 mg/min (67.7% reduction), while consumption of VEH-E-gel was sustained at 9.9 mg/min (39.3% reduction), producing a significant effect by THC to modify gelatin consumption (*Figure 2c*, two-way ANOVA, repeated measures $F_{1,14}$ = 7.604, p=0.015, Sidak's and *Figure 2—figure supplement 1b*, two-way ANOVA, $F_{1,14}$ = 6.05, p=0.03, Sidak's). By sharp contrast, mice that had consumed THC-E-gel the day prior consumed VEH-E-gel on day 3 at a significantly slower rate (6.2 mg/min) during the access period, suggesting an aversive memory to THC-E-gel (*Figure 2d*, two-way ANOVA, repeated measures $F_{1,14}$ = 4.865, p=0.045, Sidak's and *Figure 2—figure supplement 1b*). Consequently, mice exposed to THC-E-gel on day 2 significantly decreased their total VEH-E-gel consumption on days 2 and 3 (*Figure 2e*, two-way ANOVA, $F_{1,83}$ = 37.51, p<0.001, Sidak's). These data suggest that, on day 2, mice consumed high enough quantities of THC to induce a typically i.p.-high-dose (5–10 mg/kg) cannabimimetic response resulting in an avoidance to gelatin on day 3.

Reduced spontaneous locomotion is a hallmark response to THC in mice. To address whether THC-E-gel consumption impacts spontaneous locomotion, we video-recorded the traveling distance of mice during the gelatin access period (total distance in cm over 2 hr) (*Figure 2f*). *Figure 2g–i* shows that locomotion during the consumption period initially reached (1200 cm/5 min) and then steadily decreased over the 2 hr session on day 1, as expected in mice that are habituating to the environment. On day 2, spontaneous locomotor activity between mice that consumed VEH-E-gel and THC-E-gel diverged after 40 min, showing a significant decrease in total locomotion in mice that consumed THC-E-gel (*Figure 2h*, two-way ANOVA, repeated measures $F_{1,14}$ = 11.18, p=0.005, Sidak's). Thus, this reduction in locomotion parallels a corresponding reduction in consumption in *Figure 2e* that is significantly different on day 2 (*Figure 2j*, two-way ANOVA, $F_{1,42}$ = 0.3413, p=0.562, Sidak's). Importantly, spontaneous locomotion of mice exposed to VEH-E-gel and THC-E-gel was similar on day 3. Together, these results show that consumption of THC-E-gel induced hypolocomotion on day 2 after 40 min of access. Additionally, the decreased consumption of VEH-E-gel on day 3 is likely due to an aversive memory of the THC exposure period and not to hypolocomotion. Thus, E-gel incentivizes voluntary THC consumption to induce robust hypolocomotion, a hallmark cannabimimetic response, within 40 min of access.

## Consumption of THC-E-gel results in concomitant increases in the levels of THC and its metabolites in brain tissue

Several studies have shown that the PK profile of THC (5 mg/kg, i.p.) results in peak circulating concentrations of THC (1000 pmol/g), its bioactive metabolite 11-OH-THC (300 pmol/g), and its inactive metabolite 11-COOH-THC (100 pmol/g) in the brain after 2 hr (*Vozella et al., 2019*; *Torrens et al., 2020*). To determine the PK profile of THC-E-gel consumption (10 mg/15 ml) and considering the hypolocomotion behavior occurring during the consumption window, we collected plasma and brain tissue samples after 1 hr of consumption, at the end of the 2 hr access period, as well as 30 min (2.5 hr) and 24 hr (26 hr) following the 2 hr access period (*Figure 3a*). *Figure 3b* shows THC levels in the brain reached 500–600 pmol/g tissue between 1 hr and post 2.5 hr time point and was below 50 pmol/g tissue after 24 hr. Remarkably, 11-OH-THC and 11-COOH-THC levels in the brain increased concomitantly to THC levels, reaching 400–500 pmol/g tissue and 200–350 pmol/g tissue, respectively, between 1 hr and the post 2.5 hr time point, and both were also below 50 pmol/g tissue after 24 hr. Thus, levels of both $CB_1R$ agonists, THC, and 11-OH-THC, concomitantly peaked after 1 hr of THC-E-gel consumption, a result that matches the hypolocomotion response measured starting at 40 min during the 2 hr consumption period (*Figure 1*). Furthermore, THC and 11-OH-THC levels in brain tissue were lower, near zero, 24 hr after the 2 hr exposure, as previously reported (*Kreuz and*

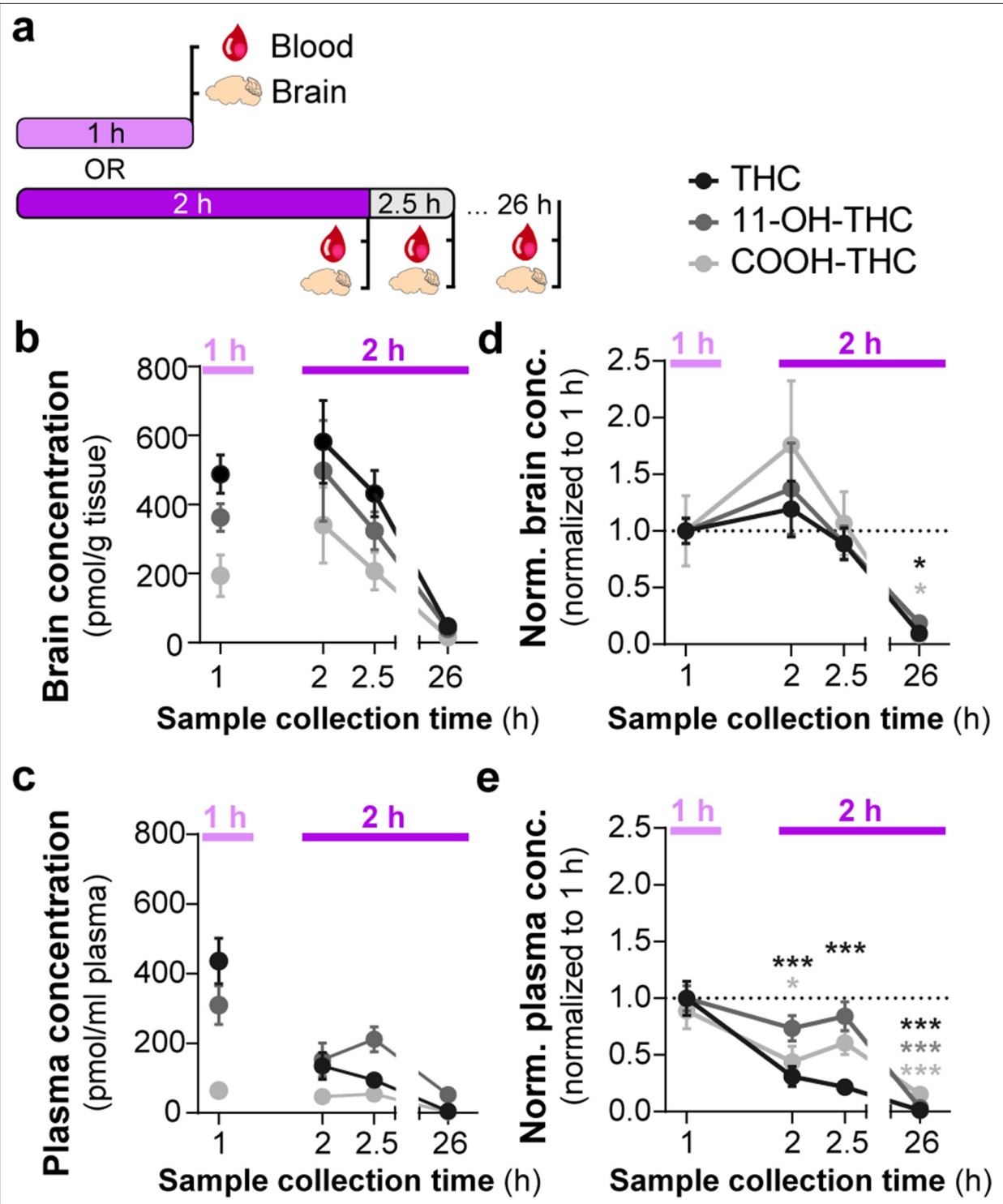

**Figure 3.** Consumption of Δ⁹-tetrahydrocannabinol (THC)-E-gel results in concomitant increases in the levels of THC and its metabolites in brain tissue. (**a**) Diagram outlining gelatin exposure paradigm where blood and brain samples were collected immediately following 1 hr and at 2, 2.5, and 26 hr from the beginning of 2 hr access to 10 mg/15 ml THC-E-gel. (**b**) Brain concentration of THC, 11-OH-THC, and COOH-THC after E-gel exposure, 1 hr access is separated due to a reduced total access time to THC-E-gel compared to the other time points. (**c**) Plasma concentrations for the three compounds plotted similarly to (**b**). (**d, e**) PK concentrations in brain (**d**) and plasma (**e**) normalized to the 1 hr access period. Statistical comparison to 1 hr two-way ANOVA, Sidak's, *p<0.05, **p<0.01, and ***p<0.001, N = 8–15.

The online version of this article includes the following source data for figure 3:

*Figure 3 continued on next page*

*Figure 3 continued*

**Source data 1.** Brain tissue concentrations of Δ⁹-tetrahydrocannabinol (THC) and metabolites by sex.

**Source data 2.** Plasma concentrations of Δ⁹-tetrahydrocannabinol (THC) and metabolites by sex.

*Axelrod, 1973*; *Johansson et al., 1989*). THC levels in plasma reached approximately 400 pmol/g tissue at the 1 hr time point and decreased thereafter (*Figure 3c*). Statistical comparisons between the 1 hr and 2 hr exposure periods were limited due to different treatment paradigms, prompting the normalization of all PK values to the 1 hr exposure period samples (*Figure 3d and e*). Brain samples were all increased at 2 hr relative to 1 hr exposure but significant differences to 1 hr exposure was only found at the 26 hr collection time point (*Figure 3d*, one-way ANOVA, $F_{3,139}$ = 14.03, p<0.001, Sidak's). Alternatively, plasma samples were significantly decreased at 2 hr for THC and COOH-THC while all three compounds were significantly decreased at the 26 hr collection time point (*Figure 3d*, one-way ANOVA, $F_{3,141}$ = 35.23, p<0.001, Sidak's). Correlation of PK findings with cannabimimetic triad results did not reveal any significant relationships (*Figure 3—source data 1 and 2*). Note that 11-OH-THC and 11-COOH-THC levels peaked after 2 hr of consumption, which contrasts with the early-onset hypolocomotion response measured in *Figure 1* after 40 min of gelatin access. Thus, PK analysis of high-concentration THC-E-gel consumption demonstrates parallel accumulation of THC and 11-OH-THC in the brain, a unique profile that differs compared to previously established PK profile resulting from THC-i.p. injection (*Vozella et al., 2019*; *Torrens et al., 2020*).

## Correlating i.p. THC and THC-E-gel triad cannabimimetic responses predicts THC-E-gel-dependent behaviors

To further establish the pharmacological relationship between i.p. THC injections and THC-E-gel consumption after 1 hr and 2 hr consumption along with the low variability in the cannabimimetic responses triggered by both routes of administrations, we calculated 'predicted THC doses" by correlating their cannabimimetic responses across experiments (*Figure 4a*). Thus, we extrapolated the relative i.p. dose for each cannabimimetic response triggered by consumption by plotting the cannabimimetic response following consumption onto the dose–response curve of THC-i.p. as reference (*Figure 4b–d*). *Figure 4b–d* also shows that 1 hr access to high-concentration THC-E-gel triggered greater cannabimimetic responses compared to 2 hr access. Consequently, this resulted in a higher 'predicted i.p. dose' shown by dotted lines tracked to the i.p. dose–response curves. Of note, 1 hr access to high-dose THC-E-gel triggered stronger hypolocomotion and reduction in core body temperature corresponding to 10.3 and 11.6 mg/kg THC i.p., respectively, and analgesia corresponding to 4.5 mg/kg THC i.p. (*Figure 4b–d*). By contrast, 2 hr access to high-concentration

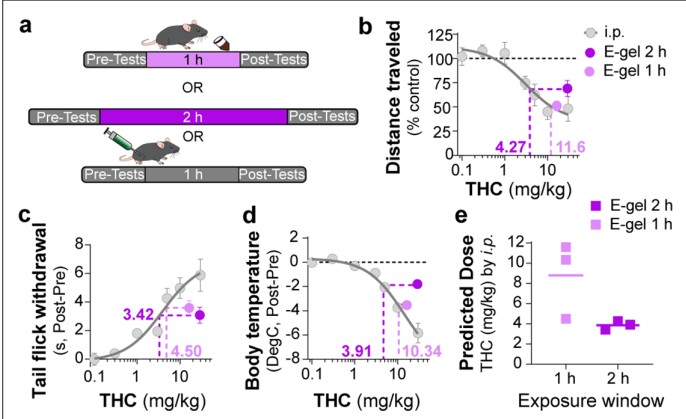

**Figure 4.** Correlating intraperitoneal (i.p.) Δ⁹-tetrahydrocannabinol (THC) and THC-E-gel triad cannabimimetic responses predicts THC-E-gel-dependent behaviors. (**a**) Diagram of 1 hr and 2 hr THC-E-gel exposure and i.p. administration with behavioral tests. (**b–d**) Cannabimimetic responses after THC administration by i.p. and subsequent dose–response curve in gray. Responses after 1 hr or 2 hr exposure to 10 mg THC-E-gel are plotted with dotted lines tracking to relative THC-i.p. dose–response. (**e**) Predicted i.p. dose after 1 hr and 2 hr THC-E-gel exposure window from all three triad behaviors.

THC-E-gel triggered a comparable response in the three cannabimimetic behaviors corresponding to 3–4 mg/kg THC i.p. (*Figure 4b–d*). *Figure 4e* illustrates the predictive value of these calculations, and the larger variability for the 1 hr access predicted dose of 8.8 ± 2.2 mg/kg i.p. and 3.7 ± 0.3 mg/kg i.p. for 2 hr access, a 2.4-fold higher predicted dose after 1 hr access. The variability between the cannabimimetic response for the 1 hr access results suggests a difference in the PK profile of THC at 1 hr compared to 2 hr access (see *Figure 3*). Together, these results indicate that consumption of high-dose THC-E-gel triggers strong cannabimimetic responses, comparable to i.p. injections of THC between 4 and 12 mg/kg, although this is not necessarily adaptable to all behavioral readouts.

## THC-E-gel consumption and THC i.p. injections induce sex-dependent changes in acoustic startle responses

Acoustic startle responses in mice represent a well-established preclinical approach to evaluate an unconditional reflex characterized by the rapid contraction of muscles to a sudden and intense startling stimulus. It is an especially useful measure in preclinical research as it is consistent across species and involves simple neural circuitry in sensorimotor gating (*Pantoni et al., 2020*). It is known that i.p.

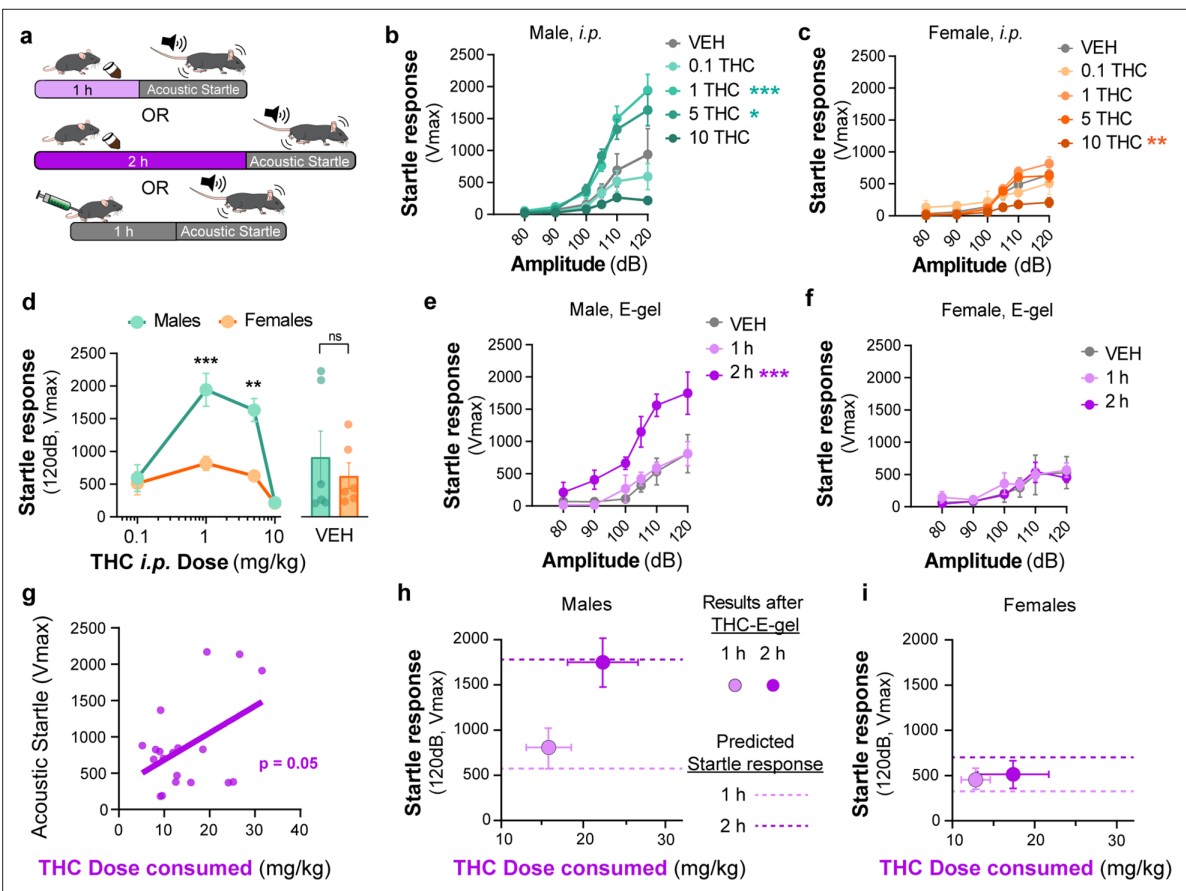

**Figure 5.** Sex-dependent acoustic startle responses after intraperitoneal (i.p.) injection of $\Delta^9$-tetrahydrocannabinol (THC) and high-concentration THC-E-gel consumption. (**a**) Diagram of THC-E-gel exposure or i.p. administration followed by acoustic startle response behavioral testing. (**b, c**) Male and female acoustic startle responses after i.p. administration of THC in response to escalating tones (80, 90, 100, 105, 110, and 120 dB) following i.p. administration of THC in males (**b**) and females (**c**). (**d**) Male and female acoustic startle dose–responses to a 120 dB tone after i.p. THC administration. Results are mean ± SEM. one-way ANOVA, Sidak's comparing vehicle (VEH) and i.p. THC dose between males and females, **$p<0.01$, ***$p<0.001$, N = 6–11. (**e, f**) Male and female acoustic startle responses after 1 hr or 2 hr THC E-gel exposure in response to escalating tones (80, 90, 100, 105, 110, and 120 dB). (**g**) THC dose consumption based on grams consumed and individual body weight correlated with individual acoustic startle response after 2 hr exposure. (**h, i**) Startle response to a 120 dB tone for males (**h**) and females (**i**) after 1 hr or 2 hr access to THC E-gel. Predicted doses calculated from a second-order polynomial of i.p. dose–responses are plotted to show the consistency in predicted dose–response after E-gel exposure.

The online version of this article includes the following figure supplement(s) for figure 5:

**Figure supplement 1.** Methodology for $\Delta^9$-tetrahydrocannabinol (THC)--E-gel prediction of a behavioral response.

injection of THC (6 and 10 mg/kg) reduces acoustic startle in male mice (**Long et al., 2010**; **Nagai et al., 2006**; **Tournier and Ginovart, 2014**). Thus, whether acute startle response is affected in a sex-dependent manner and by lower dose THC delivered i.p. or via oral consumption remains unknown. Here, we compared the effect of THC-E-gel consumption and THC i.p. injections on acute acoustic startle in male and female mice. Acute startle responses were measured as the peak velocity of startles ($V_{max}$) using an accelerometer and following audible tones of 80, 90, 100, 105, 110, and 120 dB delivered either 1 hr after i.p. administration of THC (from 0.1 to 10 mg/kg) or immediately after access to THC-E-gel (10 mg/15 ml) (**Figure 5a**). THC administration via i.p. injection induced a significantly increased startle response in males at 1 and 5 mg/kg with a trend decrease at 10 mg/kg, and only a significantly decreased startle response in females at 10 mg/kg (**Figure 5b and c**, males: v. VEH two-way ANOVA $F_{4,122} = 13.89$, p<0.001, Sidak's; females: v. VEH two-way ANOVA $F_{4,116} = 6.76$, p<0.001, Sidak's). **Figure 5d** shows the male and female startle responses that occurred at the 120 dB tone exposures and follows an inverted U shape characterized by (1) increased acute startle response at 1 and 5 mg THC (i.p.) in males, (2) absence of such response in females, and (3) a comparable reduction in acute startle response in both males and females at 10 mg THC (i.p.) in males and females (two-way ANOVA $F_{3,23} = 26.66$, p<0.001, Sidak's).

THC-E-gel consumption by males and females also triggered a sex-dependent startle response. **Figure 5e and f** shows that only males that were allowed access for 2 hr to THC-E-gel exhibited an increase in acute startle response (2.2-fold increase) (males: two-way ANOVA $F_{2,70} = 26.85$, p<0.001, Sidak's). Further analysis of the relationship between THC-E-gel consumption and modification of the acute startle response delivered at 120 dB resulted in a significant correlation (**Figure 5g**, p=0.05).

Finally, we sought to determine whether the predicted dose calculations from **Figure 4e** could be applied to the i.p. THC acoustic startle response dataset to test the accuracy and generalizability of the dose-prediction model. Specifically, we plotted the predicted doses of 3.7 and 8.8 mg/kg i.p. (from **Figure 4e**) onto **Figure 5d** depicting the acoustic startle response measured at 120 dB resulting from i.p. injections (**Figure 5—figure supplement 1a and b**). This produced predicted startle responses of 310 cm/min for 1 hr consumption and 688 cm/min for 2 hr consumption for females, as well as 558 cm/min for 1 hr consumption and 1733 cm/min for 2 hr consumption for males. We then plotted the measured acute acoustic response ($V_{max}$, 120 dB tone) following THC consumption in **Figure 5h and i**, as well as the predicted acute startle responses (dashed lines) from **Figure 5d** (**Figure 5—figure supplement 1a–c**) shows methodology. The predicted acute startle responses in males exposed to 10 mg/15 ml THC-E-gel for 1 hr and 2 hr access were close to, or within, the standard error of the measured startle response following THC-E-gel for 1 hr and 2 hr access (**Figure 5h and i**). This dose-prediction model demonstrates the reliability of voluntary THC-E-gel consumption as a behavioral paradigm to produce consistent cannabimimetic responses across different experimental modalities.

## Discussion

Here, we report a novel experimental approach that enables the behavioral impact of voluntary oral consumption of high-dose THC by adult mice. Access to E-gel for 2 hr over a 2-day period incentivizes robust consumption, and at the highest dose tested here (10 mg/15 ml), mice of both sexes consumed ~30 mg/kg THC in 2 hr on the second day. Acute consumption of THC triggers commonly established cannabimimetic responses, the potencies of which were right-shifted compared to the responses measured with i.p. injections. Furthermore, we discovered that acute consumption of 10 mg/15 ml THC-E-gel increases the acoustic startle response in males and not in females; whereas i.p. injection of THC triggers a dose-dependent, inverse U-shaped, impairment of acoustic startle response that was also more pronounced in males than females. Our study provides important translational results at two levels: voluntary consumption of THC by rodents and its sex-dependent impact on acoustic startle response as a measure of psychomotor reflexive behavior.

Mice of both sexes consumed similar amounts of VEH-CTR-gel and VEH-E-gel, and none consumed more than 20% of their daily caloric intake, indicating comparable consumption behaviors. However, consumption of high-concentration THC-CTR-gel (4 mg) was inconsistent, and 41% of the mice completely avoided consumption (as assessed by an unbroken gelatin surface at the end of the 2 hr access period) (**Figure 1c**). By contrast, consumption of THC-E-gel (10 mg, i.e., 2.5× more concentrated) was more consistent with a total consumption rate of 0.95 g/2 hr (**Figure 1d**). This difference in consumption between THC-CTR-gel (4 mg/15 ml) and THC-E-gel (10 mg/15 ml) is likely due to the

chocolate flavoring in Ensure that masks the strong odor and bitter taste of high-concentration THC and its aversive properties. At higher doses of THC-E-gel exposure, we found more variability in dose consumed (*Figure 1e*), a consumption behavior similar between sexes (*Figure 1—figure supplement 1b*). Future studies of such increased variability at even higher doses of THC may reveal differences in absorption or metabolism for example. Significantly, mice that consumed the higher-dose THC-E-gel on day 2 consumed remarkably less VEH-E-gel on day 3 (*Figure 1—figure supplement 2c*). This decrease in consumption is likely due to the development of aversive conditioned associations to higher-dose THC. Thus, the THC-E-gel experimental approach reported here also enables the study of aversive memory to voluntary oral consumption of high-dose THC.

I.p. injection of THC induces hypolocomotion, analgesia, and hypothermia in mice with different median effective doses ($ED_{50}$, 1.3, 3.9, and 14.4 mg/kg, respectively) (*Figure 2b–d*). By comparison, 1 hr access to 10 mg THC-E-gel produced cannabimimetic responses that paralleled the $ED_{50}$ of i.p. injections and are equivalent to an i.p. dose of ~9 mg/kg. Also, 1 hr access to 10 mg THC-E-gel evoked a more pronounced cannabimimetic response compared to 2 hr access, agreeing with prior studies that have shown that oral gavage increases brain peak concentration of THC 1–2 hr after administration (*Deiana et al., 2012*). Oral consumption also increases 11-OH-THC levels in the brain with comparable kinetics and concentration as THC, and the levels of both cannabinoids decrease in parallel (*Figure 3b and c*). Considering that ~600 pmol/g of THC and 11-OH-THC is roughly equivalent to 3 nM of both compounds in the brain that persists over several hours, and both activate $CB_1R$ with comparable potencies, our results suggest that the accumulation of both THC and 11-OH-THC in the brain might contribute to cannabimimetic responses (*Dinis-Oliveira, 2016*). Because of the experimental design implemented for this study, we did not determine whether the variation in the time course of distinct cannabimimetic response was different following consumption. Thus, future studies that prioritize behavioral responses at multiple time points following consumption of THC using E-gel results might reveal differences in the dynamics of onset and decay in cannabimimetic responses.

We found that oral consumption of THC-E-gel produced a higher brain concentration of the primary metabolite 11-OH-THC in the brain compared to previously published concentrations after i.p. administration (*Torrens et al., 2020*). This suggests oral administration may modify the accumulation of 11-OH-THC or its metabolism in the brain. Both males and females were studied for all pharmacokinetic time points, and we found no significant sexual differences (*Figure 3—source data 1 and 2*). Finally, considering that voluntary oral consumption of 10 mg/15 ml THC results in nanomolar concentrations of THC and 11-OH-THC for several hours, the time-dependent reduction in cannabimimetic response that follows their maximal response may also be due either to $CB_1R$ desensitization/tolerance or to redistribution of the drug within brain parenchyma.

An i.p. injection of THC 6 and 10 mg/kg in male mice reduces acoustic startle behaviors (*Nagai et al., 2006*; *Tournier and Ginovart, 2014*). We show here that THC-i.p. induces a dose-dependent biphasic behavioral response that is more pronounced in males than females, demonstrating sex-dependent sensorimotor behaviors, and confirming that THC impacts neurocognitive function in a sex-dependent manner (*Figure 1—figure supplement 3*; *Cha et al., 2007*; *Harte and Dow-Edwards, 2010*; *Gur et al., 2012*). THC-E-gel (10 mg/15 ml) consumption also increased the response to acoustic startle preferentially in males compared to females. Whether the dose of THC formulated in E-gel can be increased to levels that remain palatable to mice and might trigger the pronounced reduced acoustic startle measured with 10 mg THC injection i.p. remains an open question.

Analysis of the behavioral responses following i.p. injection and consumption of THC-E-gel enabled us to propose a model that correlates the doses of THC capable of producing comparable behavioral responses, emphasizing the robustness of this experimental approach. Thus, the flexibility of the THC-E-gel experimental approach may extend its utility as a substitute for traditional i.p. injections, better bridging the translational gap between preclinical investigations and human use. For example, the THC-E-gel experimental model can be easily modified and implemented to measure, in a less invasive manner, the impact of oral THC consumption on additional mouse behaviors, including self-administration and preference/aversion, paradigms that require multiple treatment regimens.

In conclusion, we report a new experimental approach that achieves robust voluntary oral consumption of THC in adult mice by formulating THC in a chocolate-flavored sweetened E-gel. Given the recent rise in use of *Cannabis* products that contain high doses of THC such as edibles (*Freeman*

*et al., 2021*), this voluntary consumption model allows the study of its effect on translational relevant behaviors, including sex-dependent psychomotor reflexes in mice.

## Additional information

### Funding

| Funder | Grant reference number | Author |
|---|---|---|
| National Institute on Drug Abuse | F 31 DA055448-01 | Anthony English<br>Michael R Bruchas<br>Nephi Stella |
| National Center for Complementary and Integrative Health | R01 AT011524-01A1 | Benjamin Bruce Land |
| National Institute on Drug Abuse | R21 DA051558-02 | Anthony English<br>Fleur Uittenbogaard<br>Dennis Sarroza<br>Anna Veronica Elizabeth Slaven<br>Nephi Stella<br>Benjamin Bruce Land |
| National Institute on Drug Abuse | R37 DA033396-10 | Anthony English<br>Michael R Bruchas |
| National Institute on Drug Abuse | P30 DA048736 | Anthony English<br>Fleur Uittenbogaard<br>Michael R Bruchas<br>Nephi Stella<br>Benjamin Bruce Land |
| National Institute on Drug Abuse | P50 DA044118-01 | Alexa Torrens<br>Daniele Piomelli |

The funders had no role in study design, data collection and interpretation, or the decision to submit the work for publication.

### Author contributions

Anthony English, Conceptualization, Data curation, Formal analysis, Supervision, Funding acquisition, Validation, Investigation, Visualization, Methodology, Writing - original draft, Writing - review and editing; Fleur Uittenbogaard, Conceptualization, Data curation, Investigation; Alexa Torrens, Data curation, Formal analysis, Investigation, Methodology; Dennis Sarroza, Data curation, Methodology; Anna Veronica Elizabeth Slaven, Data curation, Investigation; Daniele Piomelli, Conceptualization, Resources, Supervision, Funding acquisition, Project administration, Writing - review and editing; Michael R Bruchas, Conceptualization, Resources, Supervision, Funding acquisition, Writing - original draft, Project administration, Writing - review and editing; Nephi Stella, Benjamin Bruce Land, Conceptualization, Resources, Supervision, Funding acquisition, Visualization, Writing - original draft, Project administration, Writing - review and editing

### Author ORCIDs

Anthony English http://orcid.org/0000-0003-4490-8654
Michael R Bruchas http://orcid.org/0000-0003-4713-7816
Nephi Stella http://orcid.org/0000-0002-4780-8360

### Ethics

This study was performed in strict accordance with recommended guidelines for care and use of laboratory animals outlined in NIH Guide for the Care and Use of Laboratory Animals. All animals were handled according to approved Institutional Animal Care and Use Committee (IACUC) protocols (#3233-09) of the University of Washington by authors trained at the University of Washington. All experimentation was performed with every effort to minimize suffering of subjects.

Reviewer #1 (Public Review): https://doi.org/10.7554/eLife.89867.3.sa1
Reviewer #2 (Public Review): https://doi.org/10.7554/eLife.89867.3.sa2
Reviewer #3 (Public Review): https://doi.org/10.7554/eLife.89867.3.sa3
Author Response https://doi.org/10.7554/eLife.89867.3.sa4

## Additional files

### Supplementary files
• MDAR checklist

### Data availability
Pharmacokinetic dataset available in supplementary data. All other data stored in Dryad repository (https://doi.org/10.5061/dryad.000000099).

The following dataset was generated:

| Author(s) | Year | Dataset title | Dataset URL | Database and Identifier |
|---|---|---|---|---|
| English A, Uittenbogaard F, Torrens A, Sarroza D, Slaven A, Piomelli D, Bruchas MR, Stella N, Land BB | 2023 | Behavioral data for "A preclinical model of THC edibles that produces high-dose cannabimimetic responses | https://doi.org/10.5061/dryad.000000099 | Dryad Digital Repository, 10.5061/dryad.000000099 |

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
