## [Editor Report · eLife assessment]

This **important** study presents the validation of an oral Δ^9^-tetrahydrocannabinol (THC) consumption mouse model utilizing highly palatable e-capsule gelatin. The results **convincingly** demonstrate that oral consumption produced THC behavioral and physiological effects, as well as measurable brain levels. The utility of the model for chronic consumption remains to be determined. The authors have clearly acknowledged limitations of their model and areas for future study and development. As the field of cannabinoid research moves toward application of routes of administration that mimic human use, these model systems will be increasingly **important** in assessing the effects of cannabinoid-based drugs.

---

## [Referee Report · Reviewer #1 (Public Review)]

Summary:

This manuscript describes the development of an oral THC consumption model in mice where THC is added to a chocolate flavored gelatin. The authors compared the effects of THC consumed in this highly palatable gelatin (termed E-gel) to THC dissolved in a less palatable gelatin (CTR-gel), and to i.p. injections of multiple doses of THC, on the classic triad of CB1R dependent behaviors (hypolocomotion, antinociception, and body temperature).

The authors found that they could achieve consumption of higher concentrations of THC in the E-gel than the CTR-gel, and that this led to larger total dose exposure and decreases in locomotor activity, antinociception, and body temperature reductions similar to 3-4 mg/kg THC when tested after 2 hour consumption and roughly 10 mg/kg if tested immediately after 1 hour consumption. The majority of THC E-gel consumption was found to occur in the first hour on the first exposure day. THC E-gel consumption was lower than VEH E-gel consumption and this persisted on a subsequent consumption day, suggesting that the animals may form a taste aversion and that THC at the dose consumed likely has aversive properties, consistent with the literature on i.p. dosing. The authors also report the pharmacokinetics in brain and plasma of THC and metabolites after 1 or 2 hour consumption, finding high levels of THC in the brain that begins to dissipate at 2.5 hours is gone 24 hours later. Finally, the authors tested THC effects on the acoustic startle response and found an inverted dose response that was more pronounced in males than females after i.p. dosing and a greater startle response in males after E-gel dosing.

Overall, the authors find that voluntary oral consumption of THC can achieve levels of intake that are consistent with the present and prior reported literature on i.p. dosing.

Strengths:

The strengths of the article include a direct comparison of voluntary oral THC consumption to noncontingent i.p. administration, the use of multiple THC doses and oral THC formulations, the inclusion of multiple assays of cannabinoid agonist effects, and the inclusion of males and females. Additional strengths include monitoring intake over 10 minute intervals and validating that effects are CB1R dependent via antagonist studies.

Weaknesses:

1. The abstract does not discuss the reduction of E-gel consumption that occurs after multiple days of exposure to the THC formulation, but rather implies that a new model for chronic oral self-administration has been developed. Given that only two days of consumption was assessed, it is not clear if the model will be useful to determine THC effects beyond the acute measures presented here. The abstract should clarify that there was evidence of reduced consumption/aversive effects with repeated exposures.

2. In the results section, the authors sometimes describe effects in terms of the concentration of gel as opposed to the dose consumed in mg/kg, which can make interpretation difficult. For example, the text describing Figure 1i states that significant effects on body temperature were achieved at 4 mg CTR-gel and 5 mg THC-gel, but were essentially equivalent doses consumed? It would be helpful to describe what average dose of THC produced effects given that consumption varied within each group of mice assigned to a particular concentration.

3. The description of the PK data in Figure 3 did not specify if sex differences were examined. Prior studies have found that males and females can exhibit stark differences in brain and plasma levels of THC and metabolites, even when behavioral effects are similar. However, this does depend on species, route, timing of tissue collection. It would be helpful to describe the PK profile of males and females separately.

4. In Figure 5, it is unclear how the predicted i.p. THC dose could be 30 mg/kg when 30 mg/kg was not tested by the i.p. route according to the figure, and if it had been it would have likely been almost zero acoustic startle, not the increased startle that was observed in the 2 hr gel group. It seems more likely that it would be equivalent to 3 mg/kg i.p. Could there be an error in the modeling, or was it based on the model used for the triad effects? This should be clarified.

---

## [Referee Report · Reviewer #2 (Public Review)]

Summary:

The work fruitfully adds to the tools to study cannabinoid action and pharmacology specifically, but also this method is applicable to other drugs, in particular, if lipophilic in nature.

Strengths:

The addition of chocolate flavor overcomes aversive reactions which are often experienced in pharmacological treatments, leading to possible caveats in the interpretation of the behavioral outcomes.

Weaknesses:

Certainly, more THC mediated behavioral outcomes could have been tested, but the work presents a proof-of-concept study to investigate acute THC treatment.

It would have been interesting if this application form is also possible for chronic treatment regimen.

---

## [Referee Report · Reviewer #3 (Public Review)]

Summary: This manuscript explores the development of a rodent voluntary oral THC consumption model. The authors use the model to demonstrate that similar effect levels of THC can be observed to what has previously been described for i.p. THC administration.

Strengths: Overall this is an interesting study with compelling data presented. There is a growing need within the field of cannabinoid research to explore more 'realistic' routes of cannabinoid administration, such as oral consumption or inhalation. The evidence presented here shows the utility of this oral administration model.

Weaknesses: The main weaknesses of the manuscript revolve around clarification of the Methods section. All of these weaknesses are described in the "Recommendations to authors" section. Revising the manuscript would account for many of these weaknesses.

---

## [Author Response]

The following is the authors’ response to the original reviews.

We thank the reviewers for their thorough reading of the manuscript and insightful comments. We have responded to both the “public review” and the “recommendations” and feel that the manuscript is now significantly strengthened.

**Public Review comments**

**Reviewer #1:**
Weaknesses:1. The abstract does not discuss the reduction of E-gel consumption that occurs after multiple days of exposure to the THC formulation, but rather implies that a new model for chronic oral self-administration has been developed. Given that only two days of consumption was assessed, it is not clear if the model will be useful to determine THC effects beyond the acute measures presented here. The abstract should clarify that there was evidence of reduced consumption/aversive effects with repeated exposures.

Thank you for your observation. We have added language to address this in the manuscript and the abstract. The model developed in the manuscript is an acute exposure model, with the intention of further chronic exposure adaptations to be developed separately (page 2, line 29).

1. In the results section, the authors sometimes describe effects in terms of the concentration of gel as opposed to the dose consumed in mg/kg, which can make interpretation difficult. For example, the text describing Figure 1i states that significant effects on body temperature were achieved at 4 mg CTR-gel and 5 mg THC-gel, but were essentially equivalent doses consumed? It would be helpful to describe what average dose of THC produced effects given that consumption varied within each group of mice assigned to a particular concentration.

We thank the reviewer for this comment and have edited our text to clarify our results. For example, this point is further emphasized by the correlation of the data in Figure1l-n showing the relationship between individual consumption and behavioral readouts (page 11, line 225-226).

1. The description of the PK data in Figure 3 did not specify if sex differences were examined. Prior studies have found that males and females can exhibit stark differences in brain and plasma levels of THC and metabolites, even when behavioral effects are similar. However, this does depend on species, route, timing of tissue collection. It would be helpful to describe the PK profile of males and females separately.

We did compare sex dependent effects and found no significant effects after THC E-gel consumption. We’ve added additional language to address this point in the discussion (Supplementary tables T1 and T2).

1. In Figure 5, it is unclear how the predicted i.p. THC dose could be 30 mg/kg when 30 mg/kg was not tested by the i.p. route according to the figure, and if it had been it would have likely been almost zero acoustic startle, not the increased startle that was observed in the 2 hr gel group. It seems more likely that it would be equivalent to 3 mg/kg i.p. Could there be an error in the modeling, or was it based on the model used for the triad effects? This should be clarified.

We apologize for the confusion created by that data, and it has now been updated for clarity. The original ~30mg/kg was not a predicted dose consumed, but rather an expected dose consumed based on individual male v. female consumption data in Supplemental Figure S1b. For clarity on the figure, we’ve instead placed dashed lines that draw attention only to the predicted startle response expected from our THC-E-gel model. We have also updated the text which hopefully makes this clearer.

**Reviewer #2:**
Weaknesses:Certainly, more THC mediated behavioral outcomes could have been tested, but the work presents a proof-of-concept study to investigate acute THC treatment.It would have been interesting if this application form is also possible for chronic treatment regimen

We agree that a chronic treatment regimen and additional behavioral outcomes is the next, most exciting step for expanding this oral THC-E-gel consumption model, and something we are actively pursuing.

**Reviewer #3:**
Weaknesses:The main weaknesses of the manuscript revolve around clarification of the Methods section. All of these weaknesses are described in the "Recommendations to authors" section. Revising the manuscript would account for many of these weaknesses.

Thank you for carefully reading through our methodology. We have made edits according to everything brought up in the recommendation section of reviewer comments.

**Recommendations for Authors**

**Reviewer #1:**
Minor edits to the text:Abstract: "intraperitoneal contingent" should be "intraperitoneal noncontingent".Line 221, this sentence needs editing for clarity.Lines 249-250, incomplete sentence.Line 284, the word "activity" is missing from "locomotor between mice".Lines 299-301, incomplete sentence.

Thank you for finding these mistakes. All these recommendations have been incorporated into the final publication.

**Reviewer #2:**
1. The typical THC tetrad includes catalepsy. Why was this behavioral outcome not monitored?

We felt that locomotion, analgesia, and body temperature were robust behavioral readouts for monitoring cannabimimetic responses and that acoustic startle served as an additional, novel means of understanding THC-E-gel effects.

1. Please specify the exact substrain of C57BL/6 (i.e., J or N or some other)

C57BL/6J mice were used for the publication. This clarification has been made in the methods section.

1. Figure S3 is not mentioned in the result part, but only in the discussion.

Figure S3 is now referenced in the main body of the Results section.

1. It might be interesting to follow up the issue that the individual THC consumption is considerable, as depicted in Fig. 1e (at high dose). This will presumably also lead to different behavioral responses. Or is there individual metabolism, also difference male vs. female?

Thank you for the suggestion. We agree that the distribution of THC doses consumed (calculation based on weight) would be worth further investigating and have now included language about this (page 20, line 436). Please note that we did not find a sex difference (Supplemental Figure S1b), but it would be exciting to discover some biologically relevant cause such as individual absorption or metabolism

**Reviewer #3:**
Major1. Methods: Were the observers of experiments blinded to animal treatment? Why or why not?

Multiple investigators performed the behavioral measurements and were not blinded to mouse treatments, but the dose consumed by each mouse remained blind. Thus, because animals consumed THC gelatin of their own volition while having ad libitum access, we performed the correlational analysis presented in Figure 1 l-n.

1. Methods: The authors could consider relating their study design to the ARRIVE guidelines and providing a statement as to whether their study adheres to these guidelines. Related to this, were mice provided with any environmental enrichment during the study?

We followed the ARRIVE guidelines with exception to investigator blinding (described above). Please note that mice were not provided with additional environmental enrichment during the study, a point that we specified in our methods (page 5, line 91).

1. Methods / Results: In the Methods it is stated that the triad of cannabimimetic behaviors was measured 1 h post-injection or immediately after gelatin exposure. Why were these timepoints chosen? Perhaps this wording should be revised because measurements of cannabimimetic effects were taken several times after drug exposure. Peak i.p. drug may occur earlier than 1 h whereas peak oral drug effect is likely to occur over a longer time period (i.e., not immediately after) due to delays of absorption and first pass metabolism. Is it possible that the authors have underestimated oral drug effects by selecting these timepoints? Please discuss.

We observed a reduction in locomotion activity starting 1 h following the beginning of exposure to the gelatin (Figure 2), suggesting initial cannabimimetic changes. Based on this observable response we chose to measure all cannabimimetic behaviors immediately following gelatin exposure. The exposure timeline for i.p. injection (1 h post-injection) was selected based on a standard published protocol (Metna-Laurent et al, 2017).

a. Pharmacodynamics: Related to this and because the aim of this paper is to establish a rodent oral dose model, could the authors discuss the need for better characterization of the time course of drug effects? For example, how might anti-nociception or locomotor activity vary following THC E-gel consumption? This is somewhat addressed in the locomotion time course in Figure 2G but could be elaborated on or discussed in more detail.

We agree that future studies should include additional time points measuring behavioral changes. This important point is now emphasized in the discussion (page 21, line 455).

b. Pharmacokinetics: Related to this point above, have the authors considered collecting blood or tissue samples from their i.p.-injected animals to assess drug pharmacokinetics as they relate to drug effect and as compared to oral THC consumption? I am not suggesting the authors conduct a completely new study for this manuscript; however, this could be raised as a future study and/or as a weakness of the current study.

We did not measure blood and tissue concentrations after i.p. administration due to the number of studies reporting these values by our co-author, Dr. Daniele Piomelli, that established these pharmacokinetic measures. Thus, we chose to reference these studies. Please note that repeating such measurements would be labor intensive, unnecessary use federal NIH resources and animals, while being very redundant to the existing literature.

c. Minor, but related to these points: In the results, page 14 line 299: the first sentence of this paragraph is confusing as written. The Reviewer recognizes that the authors are relating the pharmacokinetic work to previously published findings, but still thinks that measuring and comparing THC levels from their cohort of i.p.-injected animals would have benefitted the present study.

Thank you, this edit has been made in the manuscript.

1. Methods, Histology: The methods as described do not contain sufficient detail regarding THC and THC metabolite quantification. In addition, it is not clear from this section what Histology was performed and how (no histology results appear in the manuscript). Please add more detail to this section of the Methods.

We apologize for this typo and have corrected it in the methods section of the manuscript.

1. Methods / Results: The statistics section requires additional detail regarding the rationale for tests being performed on different datasets. In addition, a description of the curve fitting used for data in figures 1H-J, 4B-D, and S4 would be helpful to the reader.

Thank you, we have updated and provided more information regarding the curve fitting that was used in the methods and results section for the respective figure panels (page 9, line 183-184).

Minor1. Throughout: The use of the phrase "high" dose is somewhat arbitrary and not defined relative to other doses of the THC formulation throughout the manuscript. The Reviewer suggests simply stating that THC was used, specifying the dose, or justifying in the Abstract and/or Introduction the classification of "high" based on relevant literature.

Thank you for the observation. We have removed this ambiguity by specifically mentioning the dose that was consumed (e.g., abstract page 2, line 20).

1. Abstract: define "CB1" in the abstract. Although this is a common abbreviation within the field, its use should be defined.

We have added this definition in the abstract for clarification.

1. Figure 2: why are the consumption panels B, C, and D given separate labels but the locomotor data are all labeled together as panel G?

Thank you for the observation, we have adjusted the labeling, so it is equal for both sets of panels.